# Understanding Animal-Plant-Parasite Interactions to Improve the Management of Gastrointestinal Nematodes in Grazing Ruminants

**DOI:** 10.3390/pathogens12040531

**Published:** 2023-03-29

**Authors:** Patrizia Ana Bricarello, Cibele Longo, Raquel Abdallah da Rocha, Maria José Hötzel

**Affiliations:** 1Laboratório de Parasitologia Animal, Departamento de Zootecnia e Desenvolvimento Rural, Centro de Ciências Agrárias, Universidade Federal de Santa Catarina, Florianópolis 88034-001, Brazil; 2Laboratório de Etologia Aplicada e Bem-Estar Animal, Departamento de Zootecnia e Desenvolvimento Rural, Centro de Ciências Agrárias, Universidade Federal de Santa Catarina, Florianópolis 88034-001, Brazil; 3Departamento de Zootecnia, Programa de Pós-Graduação em Zootecnia, Universidade Estadual de Ponta Grossa, Ponta Grossa 84010-330, Brazil

**Keywords:** management, nutrition, One Health, parasitism, ruminants, sustainability

## Abstract

Grazing systems have great potential to promote animal welfare by allowing animals to express natural behaviours, but they also present risks to the animals. Diseases caused by gastrointestinal nematodes are some of the most important causes of poor ruminant health and welfare in grazing systems and cause important economic losses. Reduced growth, health, reproduction and fitness, and negative affective states that indicate suffering are some of the negative effects on welfare in animals infected by gastrointestinal nematode parasitism. Conventional forms of control are based on anthelmintics, but their growing inefficiency due to resistance to many drugs, their potential for contamination of soil and products, and negative public opinion indicate an urgency to seek alternatives. We can learn to deal with these challenges by observing biological aspects of the parasite and the host’s behaviour to develop managements that have a multidimensional view that vary in time and space. Improving animal welfare in the context of the parasitic challenge in grazing systems should be seen as a priority to ensure the sustainability of livestock production. Among the measures to control gastrointestinal nematodes and increase animal welfare in grazing systems are the management and decontamination of pastures, offering multispecies pastures, and grazing strategies such as co-grazing with other species that have different grazing behaviours, rotational grazing with short grazing periods, and improved nutrition. Genetic selection to improve herd or flock parasite resistance to gastrointestinal nematode infection may also be incorporated into a holistic control plan, aiming at a substantial reduction in the use of anthelmintics and endectocides to make grazing systems more sustainable.

## 1. Introduction

Parasitism is an infectious disease caused by external agents such as ticks, flies, lice, helminths, and protozoa. Together with predators, parasitism is a main threat to herbivore survival and reproduction in the natural environment [1,2]. Additionally, gastrointestinal parasites can affect animal production, health and wellbeing by causing discomfort, pain, suffering, subclinical and clinical disease, and, in extreme and/or prolonged cases, death [3,4], outcomes that lead to significant economic losses in grazing systems [5,6].

The control of gastrointestinal parasitism in many grazing farming systems depends on a complex interaction of human, animal and environmental factors. Parasite management can be done through the use of anthelmintics, grazing management and other preventive procedures, or a combination of all. The question is whether it is possible to achieve efficient control with anthelmintics while maintaining animal health and wellbeing. The use of chemical drugs to control parasitism has become increasingly inefficient, due to anthelmintic resistance developed by most parasite species in herbivores throughout the world [7,8,9,10]. For example, a survey of parasite control practices in Norway found that 80% of sheep producers used anthelmintics without knowing the parasitological status of the herd [11]. Only 11% performed diagnostic tests to support their decision-making for the use of anthelmintics. These findings are alarming and are a sample of what occurs in other parts of the world; i.e., control of parasites based on the assessment of faecal egg counts (FEC) is not adopted by producers as a routine practice [12,13]. The use of helminth diagnostic tests should be integrated in economic evaluation frameworks for improved decision-making for cattle [14]. Perhaps, if it were, the use of anthelmintics could be reduced and consequently the problem of anthelmintic resistance minimised, as well as residues in animal products and in the environment reduced [15].

Most anthelmintics used in livestock are classified as having high impact on the agroecosystems, depending on how they are used. The maximum residue excretion period is more transient in sheep than in cattle manure, but low levels of excretion may continue for long periods, extending the sub-lethal effects of drugs on the environment [16]. The degradation of faeces is essential to favour the recycling of nutrients in the soil and to reduce pathogens, including the nematodes’ third larval stage (L3) [17,18]. However, the intensive use of anthelmintics may affect the fauna of the dung that are responsible for its degradation in the environment, given that some molecules, especially of the lactone group, have been proven to be highly toxic for dung beetles [19]. An important fact is that antiparasitic chemicals such as macrocyclic lactones are excreted through urine and faeces, interfering with the survival and reproduction of pollinators, earthworms and beetles, and in the decomposition of faeces [16,20,21,22]. Therefore, the frequent use of endectocides increases the negative effects on the pasture and reduces microorganisms with biological control potential. Animal manure is one of the main ways in which these veterinary compounds spread through ecosystems [16,20,23,24]. Once in the environment, they can be transported and distributed in water or soil [25,26]. These compounds can remain in the environment, causing direct or indirect impacts on non-target organisms, such as soil invertebrates, which are known to play an important role in transforming faecal material and maintaining soil quality, being the main drivers in providing various ecosystem services [16,20,27].

Given the problems caused by resistance to anthelmintics and concerns with residues in the environment and in the products [28,29,30], new control methods are sought. Indeed, the preventive use of anthelmintics has been banned in agroecological systems, where helminths are considered a major concern [31,32]. These concerns also justify the growing consumer demand for more natural and animal-friendly systems throughout the world [28,33,34]. The existing connections between humans, animals and the environment are indisputable, and the effects of pesticides are cumulative at the different trophic levels. The One Health approach shows that it is necessary to establish agricultural models with fewer negative externalities and that promote the well-being of all living beings [35].

Additionally, animal welfare is increasingly recognized as an essential component of the social sustainability of animal production systems, with many arguing that, for production systems to be socially sustainable, societal values must be integrated [36,37,38]. Fraser et al. [39] argued that, regarding animal welfare, these values can be captured in three aspects of the animals: good physical health and biological functioning, the ability to live reasonably natural lives consistent with their evolutionary history, and positive affective or psychological states. Consumers’ preferences for outdoor, organic and grazing systems [28,33,40] are in part associated with perceptions that pasture is a more natural environment and that it results in less stressed or happier animals, as well as healthier animal food products [41]. However, although grazing systems have a great potential to promote welfare, domestic animals face risks, including lack of or poor access to food, shade and water, grazing in uneven or rocky areas, exposure to extreme climatic conditions, predation, and parasitism [2,42,43]. Nevertheless, citizens expect animals not only to be raised outdoors, but also healthy and well cared-for in these environments and systems [44,45]. Furthermore, consumers who value systems with healthy animals also tend to oppose production systems that rely heavily on pesticides, antibiotics, and hormones [28,33,46]. In this context, aspects of grazing systems that may improve animal welfare gain relevance, among them the management of parasitism.

Thus, grazing systems should aim to have animals with low worm burdens posed by gastrointestinal parasite infections. Moreover, the focus of parasitism management must be on prevention, and treatments must be as natural as possible. The development of such management tools may benefit from knowledge of some aspects of the biology of the host and the parasite, the behaviours of herbivores grazing in pastures containing parasites in natural environments and those managed by humans, and the complex interplay among the host, the parasites, and the environment. Here we aim to discuss gastrointestinal nematode parasitism as a health and welfare challenge for grazing animals and review the animal-plant-parasite interactions that may help develop alternative ways to manage parasitism and to reduce the use of anthelmintics drugs.

## 2. Effects of Gastrointestinal Nematode Infections on Biological Functioning, Behaviour, and Affective States

Ingestion of L3 through food is the most common route of endoparasite infections in grazing ruminants. The eggs of adult parasites are released into the faeces and the larvae that develop in the dung migrate to the grass, where they are ingested as L3. The pathophysiology of the infection is directly related to the parasites involved and the organs affected, with greater or lesser impacts on animal health and well-being [32]. In the case of gastrointestinal nematodes, histopathological [47,48], biochemical, haematological, and immunological [49,50,51,52] changes may follow. These changes can affect the health, appetite, body growth, body condition and digestive and reproductive functions, influence wool production, and cause pain [4,48,53,54].

Parasites of the genus *Haemonchus* can cause severe clinical conditions with serious anaemia, especially in young animals and pregnant and lactating females, a significant challenge in sheep production systems [50,55,56]. Helminths such as *Trichostrongylus colubriformis* can cause enteritis with erosion in the duodenal mucosa epithelium [48], while *Haemonchus contortus* can produce gastritis; both can cause inappetence in small ruminants. Enteritis causes pain and colic due to the mucosa infection; the local response of the inflammatory mediators involves redness, heat swelling, and pain [57]. The exact causes of the lack of appetite are unknown, but some hypotheses have been raised, such as the role of abdominal pain at the site of infection, caused by the action of parasites in the gastrointestinal tract [58]; regulation of appetite through hormones such as gastrin and cholecystokinin, which could be in altered amounts be due to the presence of parasites [59]; changes in amino acid availability; changes in digestion flow and pH; or direct neural effects on the central nervous system. Increased protein and amino acid loss via the gastrointestinal tract will reduce the amount available for other tissues [60]. Infection induces protein deficiency by increasing the demand for amino acids in the alimentary tract while reducing their supply through depression of appetite [61].

When gastrointestinal parasite infections are chronic and subclinical, symptoms such as loss in appetite, apathy, prostration and submandibular oedema may not be observed [49]. In this type of condition, infected animals may manifest a reduction in productivity, feed conversion, and growth rate. The disease may become clinically evident when the worm burden increases or nutritional status is poor [56,62]. However, subclinical infections in domestic and wild ruminants may also be expressed in the form of depressed growth and reproduction, with consequences for their survival, though these effects are often overlooked [63,64]. In young animals, subclinical parasitism can also influence bone metabolism through changes in phosphorus and calcium absorption and retention, leading to reduced bone growth [65,66] and significant reduction in weight gain and wool and milk production [5,67]. The high nutrient demand for the development and maintenance of the immune response and the repair of damaged tissues during GIN infections also contributes to the low performance of infected animals [48]. An examination of the immunological response during subclinical haemonchosis in goats indicates a predominance of the Th2 immune response [64], which has also been observed in cattle resistant to Cooperia, with the gene expression of both TH2 cytokines (IL-4 and IL-13) being detected [52]. Some of the effects of parasitism discussed above may be described as sickness behaviour, a set of nonspecific symptoms that include fever, weakness, inappetence, and malaise [68]. Anorexia or loss of appetite, however, is modulated by social factors [69], which may be a reason it is not always manifested in parasitized animals. Animals with subclinical infection may not show visible changes in behaviour, especially as prey species are unlikely to show pronounced behavioural responses to pain unless injuries are advanced [70]. But subtle changes in locomotor activity can be detected [54,71,72,73]. Changes in behaviour in parasitized sheep have been shown in the form of a reduction in the complexity of activity patterns [74], changes in lying time, or changes in demeanour, detected using Qualitative Behaviour Assessment [75].

## 3. Interactions between Nutrition and Gastrointestinal Nematodes and Animal Strategies to Cope with Parasite Infections

The interaction between parasitism and nutrition can be considered from two interrelated aspects: first, the influence of parasites on host metabolism, and second, the effect of host nutrition on nematode populations and the ability of the host to resist the pathophysiological effects of infection [76].

Reduced voluntary consumption, a feature of gastrointestinal helminthic infections, may vary from decrease in consumption in cases of chronic infection to complete anorexia in cases of acute disease [76,77]. The reduction in voluntary consumption is probably the most important factor contributing to the reduction in productivity among parasitized animals [66,78]. In moderate or severe infections, food intake may be reduced by up to 20% or more [58]. Altered protein metabolism is the main effect of gastrointestinal parasites in the animal organism, as it causes a deviation of the protein synthesis of muscles and bones in order to repair damage to the intestinal wall, produce mucus, and replace blood and plasma losses [79].

In addition to providing nutrients for maintenance, growth and reproduction, nutrition of the host can affect the infectious ability of pathogens, detrimentally alter the environment in which they reside, and improve host resistance to pathogens [57,80]. Several experiments have shown the benefits of protein supplementation as regards resistance to gastrointestinal nematode infections in small ruminants [81], in periparturient ewes [82,83,84,85,86], in cattle [87,88] and in goats [89,90]. Moreover, nutritional strategies with metabolizable protein supplementation and anti-parasitic plant secondary metabolites showed a potential to additively improve host resilience and reduce reliance on anthelmintics [91]. Despite the well-known increased demand for protein in parasitized animals, lambs prioritised the ingestion of energy-dense over protein-dense foods or medicinal condensed tannins when challenged by gastrointestinal parasitism [92].

Grazing animals are always making trade-offs between acquiring their nutritional needs and maintaining their health [93]. The choice of feeding strategy may be altered by the animal’s internal environment, making it less hospitable to parasites [94], and the preference for a given food is influenced primarily by infection level and diet composition, alongside genetic [95] and environmental factors [96,97,98,99]. Animal strategies to prevent and combat parasitism involve parasite avoidance, controlling the level of exposure to infected sites (in order to stimulate the animals’ immune system); alteration of behaviour (especially in sick animals); selection of resistant sexual partners; and protecting and assisting sick animals and offspring [100,101]. As herbivores develop strategies to combat the risk of parasitism or fight parasites, they must make complex daily decisions on foraging to maximise the intake rate and nutritional quality, while minimising the risks of parasitic infections or overcoming them. In summary, when coping with parasitic infection through foraging, herbivores focus on three strategies: preventing infection, resisting parasitism, or self-medicating [93].

The ability to avoid helminthic infections is an important prevention strategy for animal survival, and it is a condition passed on to the offspring by natural selection. Food avoidance and preferences are influenced by species, breed, and physiological stage of the animal or the parasite [95,102,103,104], characteristics of the sward [105,106], and environmental characteristics [96,97,98,107,108,109]. Ruminants select forage or preferred grazing patches based on the odour, taste [110], colour, or canopy height [2,111,112]. The preferred patches have high nutritional quality, but are usually closer to faeces and potentially very contaminated with L3. This poses a conflict to the animal between obtaining better nutrition and avoiding faeces [111,113]. The decision in favour of these nutrient-rich sites can improve animal nutrition and consequently immune response, while posing a risk of parasite contact and infection. Therefore, the animals are always making trade-offs between meeting nutritional needs or maintaining health.

Herbivore trade-off theory is based on the creation of mosaics in the pastures, which become heterogeneous landscapes [93,104]. The avoidance of certain grazing patches makes them taller and more attractive, generating a mosaic of more and less nutritious areas. These nourishing patches represent a motivational situation for the animal between meeting their nutritional needs and risking contamination, or avoiding the parasites [93,113]. In both cases the animal is seeking to prevent or defend itself from endoparasitism. However, few studies have highlighted how these choices vary in time and space [114]. The trade-off value varies over time within the forage-plant development cycle. Animals may take the risk of eating young plants and reject areas with older plants with lower nutritional quality as they continue their phenological cycle.

During the evolutionary cycle of the parasite, host herbivores are challenged to make choices between parasitic (or predatory) risk and good nutrition or investing in their nutrition and immune response [2,115]. The first trade-off situation is posed to the animal with the contact between the animal and the faeces-contaminated plant. Sheep are known to be able to make complex grazing decisions (nutrition versus faecal avoidance) based on their physiological state [111,116,117]. This dilemma, proposed in the late 1990s [102,117], was the starting point for understanding the behaviour of the grazing animals and parasitism in the pasture environment. After approximately two decades, understanding the route of transmission in the light of foraging theory based on ingestive-digestive decisions, on the diverse diets at temporal and spatial variation, on parasites’ characteristics and on animal behaviour and welfare, brought a new view regarding the control of parasites and diseases more naturally and effectively [103,116,118]. According to Lozano [101], herbivores utilise several methods for preventing parasitism, including avoiding foods that may be sources of parasite infections, selecting diets that can improve immune response, and engaging in self-medication by ingesting foods with specific antiparasitic compounds that can kill and/or expel established parasites.

With parasites that have rapid pasture development times, faecal avoidance decreases the risk of infection, as the host is less likely to graze in places with a high concentration of a very active infective larval population [116]. This is the case, for example, of the cycle of the endoparasite *H. contortus*. Animals that already have mature worms will lay eggs on day 1, and these eggs may hatch and have L3 on day 4 [119]. Thus, contact with fresher faeces may pose a threat to the animal.

Once the animal is parasitized, it may either alter its ingestive behaviour or develop anorexia in order to impede larval entry into the body [111]. In the first case, the animal chooses what to eat to improve its nutritional state in order to mount an immune response [57,120], or to self-medicate [121,122]. When adopting the strategy of avoidance behaviour, sick animals seek to reduce ingestion of L3 by reducing consumption or by being more selective [111]. Faecal avoidance behaviour and anorexia are most beneficial when the host has limited ability to mount an immune response against parasites, as is the case of young animals, and when there are parasites that have rapid development [116]. Grazing time and the amount of forage consumed within 24 h are reduced in parasitized ruminants. It is important to note that the avoidance phenomenon may be compromised in intensive grazing systems such as strip grazing.

Once the animal is sick, it can use the self-medication tool. Self-medicating behaviour has been described in humans, primates, and wild herbivorous animals [123,124], as well as in domesticated animals [122,125]. Self-medication of ruminants with parasitic tannin-containing foods has been observed experimentally and in nature [92,122,125,126,127,128,129]. Foods containing secondary compounds, especially condensed tannins, have positive effects on reducing endoparasitism [127,130,131,132]. For example, more parasitized lambs ingested tannin-supplemented feed than non-parasitized lambs [133]. In addition to the anthelmintic effect, tannins also have positive effects on the intestinal microbiota. However, it is worth noting that high levels of condensed tannins in the diet (greater than 50 g/kg DM) can cause adverse effects, such as reduced voluntary food intake [134].

Therefore, food choice studies can help better understand animal grazing behaviour and resistance to parasitism and provide tools for animal management strategies for more sustainable breeding. Solutions must include grazing management to reduce pasture larval infectivity and tactics to avoid excessive exposure to anthelmintics, including strategically targeted nutritional regimens to promote an effective immune response and minimise the pathogenic effects of worms. Understanding animal strategies and self-medication behaviour in grazing livestock may reduce the use of anthelmintics in animals and, consequently, their residues in animal products and the environment, i.e., promoting One Health and preserving the agroecosystems.

## 4. Resistance to Gastrointestinal Nematodes

The use of livestock animals resistant to nematodes may help reduce the welfare and environmental problems discussed earlier. Host resistance to infection is largely mediated by the involvement of the immune system. Hyperplasia of mast cells, eosinophil and globular mucosa and blood leukocytes, antibodies, cytokines, increased mucus production and the presence of inhibitory substances in mucus have been consistently observed in relation to the development of immunity to gastrointestinal nematodes in ruminants [47,50,51,52,135]. The association between different immunoglobulin isotypes, including IgA, IgE, IgG, and IgM, and resistance to GIN has been extensively studied in sheep. A systematic review demonstrated the findings on immunoglobulin response to GIN in the literature published up to 2019 and discussed the potential to use immunoglobulins as biomarkers [136]. In cattle, some studies have shown that immunoglobulins are markers for nematode resistance [51,52,135].

Resistance to infections may be associated with the host breed or characteristics of the individual. Resistant sheep and cattle have adopted avoidance strategies that lead to nutritional disadvantages [95,112]. Genetically resistant sheep and cattle can avoid parasites, either by choosing grazing sites or by becoming more selective [104,112,137]. Sheep of the breed Scottish Greyface that bear twins risk more in trade-off situations between obtaining better nutrition and avoiding parasite ingestion [113]. Ewes become more susceptible to infections by gastrointestinal nematodes in the peripartum period, releasing a large number of eggs in the faeces. Parasite resistance also influence the intensity of this phenomenon, often referred to as periparturient immunity relaxation [55,138].

## 5. Some Environmental and Grazing Management Effects on the Parasite-Host Cycle in the Pasture

Environmental factors such as external climatic conditions including radiation, humidity and temperature, faeces dispersal, pasture characteristics, and grazing systems may assist or impair the host animal in parasitic control. Early studies done on simple grazing systems focused on avoidance and external factors that could affect avoidance, such as dung quantity and age [102]. Later models, studied to estimate with greater complexity the influence of grazing systems on parasitic risk, proposed the need to include variables related to edaphoclimatic factors, as well as interactions between parasites and the grazing pattern of the animal [63].

The development and spread of anthelmintic resistance in the main sheep-producing regions has led to integrated management approaches of parasites with non-chemotherapeutic strategies such as grazing management. The success of such programs is dependent on a detailed understanding of environmental influences on the free-living stages of the nematode life cycle and ecology [139,140]. Strategies for utilising pastures with low L3 contamination entail the use of stubble from pastures and hay or implementing alternate grazing with species that do not harbour the primary parasites of sheep or cattle. In agroecological production systems, it is possible to include areas for the production of food for human consumption in pastoral areas. This can help interrupt the cycle of gastrointestinal nematodes by avoiding the cultivation of foods that may serve as sources of parasite infections. These areas can also be used to cultivate vegetables and then be returned to pasture in a rotational system [141]. Additionally, including areas for grain production in integrated crop-livestock systems can further aid in reducing pasture contamination and preventing infections by gastrointestinal nematodes [141,142].

Diverse environmental conditions (dryness, rainfall, humidity, temperature, UV radiation) in each geographical region will determine a different dynamic of the free-living stages of gastrointestinal nematodes. Calves in the humid Pampa in Argentina are placed on the best pastures in order to obtain a good body mass development; however, these pastures are the source of infection for the recently weaned animals, resulting from the high L3 pasture contamination [143].

Climatic conditions may interfere with the grazing behaviour of the host animal [144], as well as with the parasite cycle [121,145,146]. For example, less intense solar radiation may increase the distance that lambs forage from the dung [127,137]. Lambs’ avoidance of dung can occur when radiation strikes a pasture with poor forage cover, making the faeces-repellent odour stronger [147]. Ingestive behaviour may also affect parasitism by changing the intensity and timing of parasitic incidence in the pasture and promoting alteration of the canopy architecture and floristic landscape [116].

Solar radiation may also cause L3 desiccation [105], which occurs, for example, in rotational systems during the paddock-resting period, when there is a lowering of forage height (natural or cropped) and greater exposure of the L3 to solar radiation [112,137]. Larval desiccation will result in a decreased pasture infectivity and, consequently, a lower-than-normal parasite burden inside the host.

In pasture systems, all contact between the herbivore and the contaminated pasture represent a chance of nematode infection [108]. Thus, less pasture contact, with the fodder in a good nutritional state, may favour the decision for better nutrition with low risk of contamination in the trade-off imposed on pasture systems. Since natural pastures have more forage species, they create heterogeneous environments in time and space and, therefore, there is greater nonlinear interrelational complexity between the heterogeneous environment and time. The heterogeneous architecture of the plant composition creates microclimates and varying nutrient distribution, which plays major roles in the selection of diet by herbivores [107,148]. These natural landscapes are composed of smaller inter-hierarchically and dynamically interconnected spaces forming mosaics [149,150]. Mixed pastures with two or more forage species (multispecies) resemble heterogeneous natural pastures.

Animal behaviour responses in heterogeneous pastures are quite varied, depending on the host animal species and the spatial and temporal disposition of the faeces [114]. In heterogeneous pastures, the vegetation and faeces are not homogeneously distributed in either the natural or any pastoral system [151] or in domestic environments [152]. Additionally, the forage and legume species may also influence the ingestion of L3 by animals, which offers a potential tool for reducing contact with the parasite. Garcia-Méndez, et al. [153] compared the vertical and horizontal migration of L3 of gastrointestinal nematodes in sheep across three legume species: white clover (*Trifolium repens*), red clover (*T. pratense*), and bird’s-foot-trefoil (*Lotus corniculatus*), finding that smaller numbers of L3 were recovered from the upper stratum of the third plant species. Others [154] showed that, depending on the season, the forage species may be either a limiting or facilitating factor of L3 vertical migration. During the autumn season, which is marked by low temperatures and high humidity, L3 migration was facilitated by aruana grass, but hindered by brachiaria and signal grass. Additionally, L3 migration was observed to be longer under high humidity conditions, as moisture can permeate the grass stems and hairs. In contrast, under low humidity conditions, such as in spring, moisture only permeates through the stems. In such instances, aruana and brachiaria facilitated the migration of L3 [154]. The spread of anthelmintic resistance in the main sheep-producing regions has led to the development of an approach for the integrated management of parasites with non-chemotherapeutic strategies, such as grazing management. The success of such programs relies on a comprehensive understanding of the environmental factors that affect the free-living stages of the nematode life cycle [140].

The type of grazing system may exert an important interference in various ways with the parasite cycle and the oral transmission route. Rotational pasture systems are well accepted and used in tropical pastures, and their benefits in managing gastrointestinal nematodes are well recognized [112,155,156]. When rotational grazing systems have multi-species grazing and variable paddock-resting periods (e.g., the Voisin System) [157], the relationships between the heterogeneous environment and time are even more dynamic [158,159,160]. However, the characteristics of so-called rotational systems vary greatly among different countries and studies [108,109,112,156,161,162]. In a temperate climate, rotational systems are mainly performed by not using summer/fall pastures reserved for young animals on the winter return (turnout) [109,156]. In Australia, agriculturalists use moderate fertilisation, moderate grazing pressure, an average of five days’ occupation time and longer rest periods [156]. In the United States, continuous systems, rotating with a fixed return period and rotating with a variable period based on forage height are used [161]. The survival of L3 on pastures in Argentina was determined by their ability to adapt to environmental conditions. The longest surviving *Ostertagia*, *Cooperia* and *Trichostrongylus* genera recorded were those adapted to the cold, temperate climate characteristic of the Pampa region. The pasture infectivity seems greatly determined by rainfall with high L3 levels in autumn–winter that decreases towards summer [143]. Similar results were obtained in summer in a coastal area in southern Brazil [112,153]. Another rotational system, cited in Sweden, consisted of separate 2 ha-paddocks with a grazing period of approximately 20 weeks [163,164]. Some of the rotational systems in Brazil and New Zealand are based on the Voisin ecological principles, which focus on short occupancy time (24 to 48 h) and a variable resting period [157].

Despite their differences, rotational systems have shown good control of gastrointestinal parasites [165] and, in general, the reason is linked to the host--parasite cycle modification [156] and to the host immune system [95,104,112,116]. However, when the resting period from grazing is insufficient, pasture contamination does not reduce significantly, as L3 can survive for several weeks or even several months in the environment [166,167,168]. Long spelling periods are required for pasture to become free of contamination by L3 of *H. contortus* in the humid subtropical climate of São Paulo state, Brazil, where pasture contamination persists for up to 294 days in spring and 182 days in summer [168]. In commercial farm conditions, prior planning involving greater diversification of the area with other plant crops can provide areas with lower risk and promote environmental sustainability. The use of nutritional tools and grazing in areas of lower contamination is feasible.

The shorter grazing period offered in some rotational systems, where there is a high stocking rate followed by zero grazing, may be one of the main mechanisms of parasitic control in these systems. For example, a study evaluating the control of gastrointestinal parasites in lambs weaned in organic systems found that the number of abomasal worms was smaller in the rotational system with variable return time than in continuous grazing and another rotational system; however, body weight gains were similar and no differences in the economic value were found among any of the grazing systems [161]. In the Voisin grazing, the height of the pasture cannot be a reference to establish the resting time of the paddock, but only the state of plant development. Grazed pastures at their optimal resting point guarantee greater productivity and pasture quality [157]. Grass is cut uniformly under this system, allowing most of the blades of grass to receive solar radiation, with the probable consequence of reducing contamination by nematode L3. Low nematode infection has been reported in the rotational system based on ecological principles, such as the Voisin system, when compared with continuous systems. For instance, cattle FECs have ranged between 0 and 1400 eggs per gram of faeces on average [112]. In this same context, when investigating and comparing animal welfare and parasitism of organic and conventional dairy farms in Italy, the researchers clearly observed no differences between parameters related to welfare and the prevalence of helminth infections in different production systems [169]. According to the same authors, preventive measures in organic systems should include the regular use of parasitological diagnosis as a tool in breeding systems, which help determine whether anthelmintic treatment is necessary or can be safely delayed or withdrawn. Rotational systems with small paddocks minimise the chance of animals coming into contact with fresh faeces, which lowers the risk of contamination, as the time the animals stay in the area is shorter than the time required for the hatched eggs to reach the L3 stage. Allowing adequate rest time for the paddock after grazing and promoting L3 desiccation through exposure to solar radiation can help prevent dung from becoming reservoirs of L3 and reduce paddock contamination [112,137,153].

Finally, co-grazing cattle with a second, non-susceptible herbivore species that have different grazing behaviours can be used to reduce parasite transmission, exploiting host specificity and potentially allowing L3 to be removed from the system [170,171,172,173,174]. Most studies have demonstrated that cross-infection between cattle and sheep nematodes is usually of little significance when these ruminants share pastures. The benefits of integrated grazing occur especially when this involves adult cattle [175,176]. The integration of different herbivore species can result not only in the reduction in environmental contamination by infective parasites, but also in more and better forage [177].

## 6. Conclusions and Final Considerations

We have reviewed evidence that gastrointestinal nematode parasitism presents a main challenge to animal welfare in grazing systems. The most clear effects are on growth, good health, reproduction and fitness, but there are also indications of suffering, which refers to experiencing unpleasant emotional states [178]. As reviewed earlier, animals parasitized by gastrointestinal nematodes may experience hunger, pain and malaise. Thus, it is clear that parasitism, if not well managed, entails animal suffering. Studies investigating the affective states of animals with clinical or subclinical parasitism may add to this understanding. Play behaviour, a widely recognized indicator of welfare, could be explored in the context of parasite infections in young animals. A decrease or cessation in play frequency is a common response when environmental conditions become challenging, negatively affecting an animal’s fitness status [179]. Therefore, a reduction in play may potentially serve as an indicator of infection. In animal welfare research, judgement bias has emerged as a promising tool to measure affect [180], with previous studies showing changes in attention and judgement biases in response to pain [181] and hunger [182], two conditions associated with endoparasite infections. As such, studying these affective states in the context of clinical or subclinical endoparasite infections could provide valuable insights.

As we have reviewed, depending on factors of the parasite, the host and the environment, grazing ruminants use different strategies to avoid or control gastrointestinal nematodes and their effects on the body. In this process, they weigh the short-term motivation to access palatable nutrition against longer-term effects on fitness; occasionally, they may self-medicate to alleviate the negative experience of being parasitized. However, depending on how grazing systems are managed, animals are deprived of using these naturally evolved behaviours to avoid or control gastrointestinal nematodes. In farming systems, blanket treatments with anthelmintics are often the main tool used by farmers to maintain or restore animal health. With a vision of a more sustainable livestock production, we suggest that knowledge of host-parasite-environment interactions needs to be improved and refined and incorporated into pasture management systems, which may restore ecological balance and reduce dependence on pharmacological tools.

Yet, although much progress has been made, there are still few studies regarding animal adaptation strategies and individual control of parasitic diseases. Importantly, there are not many studies on animal behaviour strategies regarding risk of parasitism in grazing systems in subtropical or tropical climates involving warm and humid parasites such as *Haemonchus*. Most of the identified studies were conducted in a temperate climate, in pastures with C3 type photosynthetic-metabolism plants or in natural grasslands in cold regions. Also, the studies on grazing behaviours against the threat of parasitism or its management need to be expanded to more breeds and animal categories. Most studies have used Soay sheep, Scottish Blackface, and Scottish Greyface, although some have employed Katahdin lambs. In contrast, few studies have evaluated the behaviour of tropical and subtropical animal breeds, such as Crioula Lanada [50], Santa Ines [183], and Red Massai [184], which are naturally resistant to parasitism. We also encourage studies evaluating the behavioural adaptation strategies of herbivores and domestic ruminants in rotational systems, co-grazing among herbivores in multispecies pastures and with variable resting times that, as discussed, have greater environmental complexity, which can help or challenge animals in the pasture. Given the complexity of the factors and their interactions on a space-time scale, further studies with multifactorial analyses are indicated. Meta-analyses can be important for preselecting these factors.

Reducing the use of pharmacological tools may improve the welfare of grazing ruminants by avoiding exposure to potentially aversive human-animal interactions associated with drenching management. This may also avoid or reduce a less discussed and less understood issue, which is the undesirable side effects on animals treated with anthelmintics. Strategies and parasite-management tools aiming to reduce or avoid the use of anthelmintics can contribute to promote One Health, as they integrate the health of the animals, the environment (for example, by reducing the adverse effects on biodiversity and nutrient cycling), and humans (for example, by reducing residues in the food and in the environment that may enter the food chain).

A better management of gastrointestinal nematodes and increased animal welfare in grazing systems may be achieved with the following measures: using multispecies pastures; mixing animal categories or species; using resistant breeds adapted to the local environmental conditions and its crosses [51,185]. Another important measure is to maintain adequate nutrition to maintain or boost the immune system in young animals and in stressful life events like parturition and weaning, as well as under extreme climate events that can lead to a general scarcity of feed supply [62,186]. The host-parasite specificity can be exploited to produce clean pastures or reduce contamination by the L3 of gastrointestinal nematodes when different herbivores share pastures. Mixed or multi-species pastures may favour parasite control in many ways, as they may promote better nutrition with high levels of protein, as well as offer the animals greater opportunity for self-medication with plants containing antiparasitic compounds.

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
