# Peer review of "Understanding Animal-Plant-Parasite Interactions to Improve the Management of Gastrointestinal Nematodes in Grazing Ruminants"

_pathogens, 2023, doi:10.3390/pathogens12040531_

Round 1

Reviewer 1 Report

Pathogens-2130234 (to authors)

Sustainable management of gastrointestinal nematodes in grazing ruminants

Bricarello et al

The authors presented an interesting review bring together existing and new concepts around host – parasite interactions in grazing conditions, where the link between the host and their grazing environment clearly poses trade-offs regarding the benefits of nutrient intake and the (negative) consequences of parasite exposure. As an overall comment, whilst there are quite a good number of reviews on host – parasite interactions, where nutrition often plays a role, it could perhaps be made stronger what the different take is here.

Going through the manuscript, a good few specific queries arose, and these are presented here in the usual way for the authors to consider.

L33: are parasites not pathogens? Please rephrase

L36: This sentence is not different in meaning as the previous one. Why the apparent repeat?

L36: “Occasionally” is a strange word use. I'd suggest: an in extreme and/or prolonged cases.

L40: human (no s)

L40: "environmental" is better

L47: Current situation in what sense?

L51. Strong statement (little farmer adoption of doing FEC routinely), which is likely true, but that could benefit from a reference?

L55: replace with "having" for clarity?

L60: perhaps replace with "pathogen, including nematode L3" to point towards this being a general aspect? Also, L3 is not defined yet

L66. Not clear; suggest to replace "that for of antiparasitic control" with "the preventative use of anthelmintics" for clarity.

L91. Rephrase into "below appropriate threshold". Free from parasites is Utopian thinking that displays unnecessary naivety.

L107: Why reference to "mixed" infections?

L111: rephrase, pregnancy and lactation are reproduction functions, and this part can be made clearer.

L113: rephrase to avoid the use of "severe" twice in one sentence.

L116: this suggests Trichostrongylus spp. do not show inappetence? I beg to differ so please reconsider how to phrase these examples to avoid making questionable statements.

L119. Here work from Hutchings et al 1998 (https://doi.org/10.1006/anbe.1998.0761), Kyriazakis 2010 (http://dx.doi.org/10.1016/j.anifeedsci.2010.01.001) etc can be usefully included, as the text now refers to physiological aspects that might effect intake but do not address the question of "why"?

L125: Well, this need to be considered in light of what the authors consider a definition of sub-clinical disease. Many papers have often referred to reduced appetite, loss in production efficiency etc in the absence of overt clinical signs such as profound diarrhoea and anaemia etc as subclinical situation. Please define subclinical from your perspective and use its implication consistently throughout.

L149. What is meant with "progressive decrease" in this context?

L154. Please qualify. Can you comment if this is a single observation, or are there more in support that feed composition influences anorexia? In my view, there is not a strong body of evidence so qualify this please.

L165. This is an interesting observation and deserve some more discussion as to "why" as that new data is counterintuitive to the expectation?

L171. Explain more about relation between infection level and feeding strategy.

L177. You may want to rephrase this, as this now seems to suggest an active decision process. Could you not argue that grazing strategies must he the outcome of such "considerations" but how this is being done remains elusive?

L201. What is a phenological cycle?

L204. Would you reconsider "periodically" here as parasite exposure is virtually always happening?

L214. It is not very clear what this new view regarding parasite control is. Can you spell it out? The text following is not very focused to answer that uncertainty, in my view.

L238. Is this a case of self-medication? These ewes did not have the possibility to avoid the condensed tannins. only diet selection experiments / observations can provide such evidence.

L243. Can you spell out what these "tools" are?

L251. I do not disagree this long list of parameters are associated with immune response but can a comment be made about the challenge to consider elevated such markers as a hallmark for resistance or infection? For example, does high levels of antibodies suggest high level of resistance or exposure?

L257 Prepared? please rephrase.

L260. Is the study referred to relevant to make such strong statement that seem to suggest that Scottish Greyface is considered a genetically resistant breed?

L276. Increase or decrease of solar radiation? Please endure directional effects are included when making such statements.

L308. Please expand on this impact of season on larval migration.

L332. Please comment on the trade-off between benefits of long pasture rest and insufficient nutrient supply. what can one do under commercial farming conditions?

L353. This (regular routine parasitological diagnosis) is key for the success of all these alternatives raised. bring this up stronger.

L377. Please explain what is meant by judgement bias, and who is the judger here.

L410. A comment to the effect that variation in behaviour is more subtle that say parasitological examinations may be useful here.

L424. The use of adequate nutrition is an often quoted alternative but needs to be considered in a framework of general scarce feed supply.

Author Response

Dear Editor and Reviewers,

            We gratefully acknowledge the editor and reviewers for the valuable comments, suggestions, and corrections. We revised our manuscript following your suggestions, which we believe, have significantly improved its quality. Please note that your comments are reported below. Our responses immediately follow them, indicating the changes made in the revised version of the manuscript. Please let us know if there is anything else we might still improve or clear. We are thankful for your consideration of this manuscript for publication in Pathogens.

The Authors.

Reviewer 2 Report

This paper provides a detailed review of sustainable management of grazing ruminants, including host interaction, drug resistance, pasture management, etc. However, it seems that the full text is not forward-looking enough, which needs the author to think deeply.

Author Response

(The authors gave the same response as above.)

Reviewer 3 Report

Sustainable management of gastrointestinal nematodes in grazing ruminants

The general topic of the review (from now on, the manuscript) is very interesting and deserves to be published after substantial review of its contents as some gaps in knowledge have been identified.

 General comments

The current title is wrong as it is too ambitious and does not reflect the topic covered by the manuscript; ‘Sustainable management of GIN in grazing ruminants’ is a much wider subject than the one offered by the manuscript. In the words of the authors, the manuscript deals with descriptions of “animal-plat-parasite interactions that may help develop alternative ways to manage parasitism” but not with actual sustainable ways to manage such parasitism. Therefore the current title should be narrowed down to appropriately reflect the focus of the manuscript. Suggested title: “Considerations on animal-plant-parasite interactions to help managing gastrointestinal nematodes in grazing ruminants”.

Another two main considerations are geographical reach and ruminant species. Regarding geographical reach, as a geographical region for the reviewed topics is not specified, the reader expects to find answers to the presented problematic applicable worldwide. However, the use of the bibliography oscillates between ‘internationalisation’ and a much more confined ‘nationalisation’, as in many cases only Brazilian references are used and other non-Brazilian ones are ignored. Should the authors decide to limit their review to a pure Brazilian scenario, then several references appearing in the original review should be deleted and not discussed; going this way, however, would hinder the quality of the paper.

 The same applies to the ruminant species; the authors should decide whether the manuscript covers ‘grazing ruminants’ or only sheep. The title suggests that both cattle and small ruminants (i.e. sheep and goats) are the species the manuscript deals with. However, the bibliography used refers mainly to sheep. Should then the manuscript be focused on the species only? If not, more relevant bibliography must be incorporated.

As this is a review paper, it is expected to contain and discuss all the references from the relevant literature dealing with the topics at hand. However, there are many more-than-relevant works from around the world that do not appear in the review and they should; otherwise the use of the bibliography can be perceived as skewed and arbitrary. The authors must address that. While some of the topics of the manuscript are relatively new (eg. animal welfare combined with self-medicating), other topics are not, albeit the review gives it a fresh perspective (eg. environmental effects on GIN). For the not so-new topics, there is an important body of data from ‘old’ references that the authors are strongly encouraged to consult and add accordingly. Especific examples:

- The self-medicating topic (lines 223-240) is presented mainly in sheep. What about cattle?

- The section on resistance to GIN (lines 248-265) offers bibliography on sheep. Again, what about cattle?

- Same comment on lines 299-308 and on large parts of the section “Conclusions and final considerations”.

- Epidemiological features such as egg hatching, development of L3 in faeces and survival of L3 on pasture are key to the points raised on the section ‘Some environmental and management effects…’. However, the use of bibliography is very limited on these aspects. Works by O’Connor et al (2006), Besier et al (1993), Fiel et al (2012), van Dijk et al (2009) and several others are extremely relevant but are, sadly, missing.

- Lines 48 and 63: other supporting references are needed.

- The section on co-grazing is also very poor in both, content and use of references.

- The concepts of mixed animal/categories grazing (as stated in line 423 and 427-428), and adequate nutrition (as stated in lines 424-426) are not new, therefore, the whole idea of their implementation in the control and/or prevention of GIN is not novel. If both concepts are intended to be included in the manuscript – even as ‘fresh’ ideas because they are presented mixed with other control measures, they should be properly addressed. At the moment, they are under-reviewed.

 Specific comments

- Why is ref #19 included in this review when it does not deal with parasites?

- Line 115: Correct parasite species (colubrifomis, not colubriformes)

- Lines 216-222: It is not clear why this paragraph is on the nutritional section of the manuscript.

- Line 217: define “immediately infective”, as there is no such thing regarding GIN, not even in the tropical regions.

- Lines 219-222: This statement is a very simplistic one. The fact that H. contortus L3 might be available in faeces as early as 4 days after faeces have been voided does not mean that the pasture is already infected since L3 need to migrate from dung to the surrounding grass by effect of moisture provided mainly by rain.

- Lines 226-228: This statement is all very well in situations where animals can actually select forage. In intensive grazing systems (highly practiced worldwide and compliant with the idea of outdoor healthy grazing) animals do not have the option to be selective at will. In this regard, I find that  the manuscript does not cope well with ‘real production systems’.

- Lines 230-231: “pasture development times”, there is no such thing. The development of free-living stages of GIN occurs exclusively inside the faeces and not on pasture. Once L3 migrate (or are transported) onto the pasture, there is no more development, only survival.

- Lines 236-240: The other side of the coin is that high levels (concentrations) of CT in any given fodder are not beneficial because they negatively influence palatability and the animals reject such fodder. I believe the manuscript should include something about these aspects, otherwise it conveys the idea that all diet with CT are good for ruminants harbouring GIN.

- Line 266: Shouldn’t it be “Some environmental and grazing management effects on…”

- Line 284: replace “pruned” with “cropped”.

- Lines 284-285: Larval desiccation (and, therefore, death) will NOT interfere with the parasite cycle per se. Rather, the overall effect will be a diminished larval infectivity and, consequently, a lower-than-normal parasite burden inside the host. But the actual biological cycle will not change.

- Lines 311-312: Why rotational pasture systems are mentioned only for tropical regions?

- Lines 328: as stated for lines 284-285, the host-parasite cycle is NOT disrupted but modified in intensity.

- Lines 336-337: Please use appropriate grazing vocabulary. Replace “intense occupation” with “high stocking rate” (or, at the very least, “high grazing pressure”) and replace “absence of animals” with “zero grazing”.

- Lines 338-340: First, smaller than what? (bad use of English here). Second, I find this statement misleading. While Joan Burke, James Miller and Tom Terrill found lower abomasal burdens in the animals from the rotational system with variable return time (RBH) than in continuous grazing (CB) and another rotational system (RB), body weight gains were similar and they did not detect differences in the economic value between any of the grazing systems.

- Line 355: Replace “young faeces” with “fresh faeces”.

- Line 357: “infect larval stage of helminth eggs” is not proper parasitological vocabulary. Replace with  “...required for the hatched eggs to reach the L3 stage”.

- Lines 354-357: Although the idea provided in this statement is true, the idea that short stays are safe is not accurate as it does not account for the survival of L3. What happens then when animals complete the grazing circuit and are return to the paddocks to start another grazing rotation? The egg-L3 development takes place at different rates in different sections of the faecal deposition (please, check bibliography about this as it has been extremely well described), thus, the faecal deposition becomes a larval reservoir as L3 do not abandon the faeces and migrate to pasture all at once. For how long there are L3 inside the faeces ready to migrate to pasture? That depends on climatic conditions, mainly temperature and rainfall. Sadly, none of this is contemplated in the manuscript.

Author Response

We gratefully acknowledge the editor and reviewers for the valuable comments, suggestions, and corrections. We revised our manuscript following your suggestions, which we believe, have significantly improved its quality. Please note that your comments are reported below. Our responses immediately follow them, indicating the changes made in the revised version of the manuscript. Please let us know if there is anything else we might still improve or clear. We are thankful for your consideration of this manuscript for publication in Pathogens.

The Authors.

Round 2

Reviewer 1 Report

I thank the authors for taking into consideration the comments provided and am content with the way how this has been done.

Author Response

Dear Editor and Reviewers,

            We gratefully acknowledge the editor and reviewers for the valuable comments, suggestions, and corrections. We revised our manuscript again following your suggestions. Our responses immediately follow them, indicating the changes made in the revised version. We are very delighted that have significantly improved the manuscript. We are thankful for your consideration of this review for publication in Pathogens.

The authors.

Reviewer 3 Report

pathogens-2130234

General comments

The manuscript has been greatly improved from its original version. However, it stills needs some work before is read for publication; the authors need to revise the manuscript paying special attention to the detailed comments below. Besides, the English language/grammar, especially in the newly-added text, need to be improved; the text is extremely embellished in parts, making the paragraphs unnecessary long and convoluted. Some corrections are detailed below; however, please note that there are many more to pay attention to and it is not the reviewers’ task to correct the English grammar.

Specific comments

- Line 56: Add “the” between knowing and parasitological.

- Line 65: Replace “to the” with “on the”.

- Lines 111-112: Wrong statement, rephrase (free of low worm burdens?).

- Line 127: Delete “that are present in the environment”. You have already stated that those L3 are on the grass, therefore the last past of the sentence is redundant.

- Lines 127-129: This is a case of unnecessary embellishment and repetition that makes the statement sound erroneous. This sentence should simply read “The pathophysiology of the infection is directly related to the parasites involved and the organs affected, with greater or lesser impacts on animal health and well-being”. “Parasites” must be plural in this context.

- Line 132: Delete “weight loss” as this is part of the already mentioned “body growth” being affected.

- Line 159: Add a comma (“,”) after [65,66] and delete “in the production of infected animals on”, leaving the text “significant reduction of weight gain, wool and milk yields”.

- Lines 162-165: Rephrase, bad English grammar.

- Lines 188-191: I find the statement on the degree of inappetence varying with the amount of dietary protein very wrong. Either the English grammar is not correct or the authors are lacking some understanding on the effects of dietary protein on GIN infections. The ref used certainly does not back up such statement. Besides, the cited work also deals with breed responses to parasitism, note that one of the breeds showed no difference in outputs (parasitological parameters) regardless the different protein levels provided.

- Lines 197-199: As already commented on the previous review, there is too much biased towards sheep works. Works on protein supplementation and GIN have also been published for cattle (in 2000 and 2002 at least) and goats (in 2007 and 2021 at least), and should not be ignored.

- Line 210: Correct to “…level of exposure to infected sites” (delete “risk”).

- Line 214: Delete comma after “foraging”.

- Line 216: “coping with parasitic…”.

- Lines 252-257: Rewrite paragraph. It is constructed with poor English that makes it almost impossible to understand.

- Line 273: Replace “young ewes” with “ruminants”, as the reduction in forage consumption is seen in other ruminant animal categories and species.

- Lines 274: Replace “commercial systems with pasture monocultures that do not allow for animal choice” with “intensive grazing systems such as strip grazing”. Pasture monocultures is wrongly used in here. My previous comment (“…In intensive grazing systems… animals do not have the option to be selective at will) did not refer to monocultures. Apologies to the authors if that comment was not clear enough. Intensive grazing systems refer to the routine of grazing pastures at high stocking rates on paddocks or strips and rapid movement of the animals to the next paddock/strip. This takes place on either native or cultivated, associate pastures (yet, the animals have little option to select forage because the stocking rate does not allow it). In either case, the composition of the pasture is mixed, never a monoculture. Hence, the use of “pasture monocultures” does not apply. The only time when grazing on a monoculture occurs is when animals are placed on a post-harvested cultivar area, such as maize, wheat, etc. as part of an agriculture-livestock rotation system, such as the one described in L342-345.

- Line 281: Replace “more of the food containing tannin” with “tannins-supplemented feed”.

- Line 287-288: “to reduce pasture larval infectivity” is the correct term. What does “L3 contamination/stocking rate” means?

- Line 311: Delete comma after “parasites”.

- Lines 312-313: The paper cited does not claim that the Scottish Greyface breed is genetically resistant to GIN. In fact, there seems to be no literature claiming that. Just because the animals had a low FEC does not mean that are genetically resistant to GIN. Careful must be taken with the use of bibliography!

- Lines 314-317: Why is the periparturient increase of FEC included in this section? As it reads, there is a disconnection between this paragraph and the whole section.

- Lines 338-342: Please re-write this sentence by simplifying it and proper use of English language.

- Line 362: Correct to “Larval desiccation will result in a decreased pasture infectivity…”.

- Lines 381-385: Wrong use of citation style and English grammar. It should be “For example, García-Mendez et al. compared the [VERTICAL or HORIZONTAL?] migration of L3 in three legume species, white clover (Trifolium repens), red clover (T. pratense) and bird’s-foot trefoil (Lotus coniculatus). They recovered smaller numbers of L3 from the upper stratum of the latest plant species [148]”.

- Line 385-394: Same comment. Plus, “L3 larvae” should not be used (it’s like saying ‘third-stage larvae larvae’), just “L3” after the abbreviation has been explained at first use.  The sentence should read “The forage species may be either a limiting or facilitating factor of [VERTICAL or HORIZONTAL?] larval migration, as shown by Rocha et al. During autumn, which is …, the migration was facilitated by aruana grass but hindered by brachiaria and signalgrass. Moreover, the larval migration was longer in the presence of high humidity as moisture sips through the grass stem and hairs, as opposed to low humidity conditions (e.g. in spring) when moisture sips through only the stems. In such instances… [149]”.

Plus, “L3 larvae” should not be used (it’s like saying ‘third-stage larvae larvae’), just “L3” after the abbreviation has been explained at first use. Check this throughout the whole manuscript.

- Lines 452-455: Write whole sentence, wrong use of English grammar.

- Line 465: “potentially allowing”

- Line 512: Citations needed to back up the statement of breeds naturally resistant to parasitism.

- Line 533: What is the factor that can lead to scarcity of feed supply? As it’s written, the text implies that all sort of stressful events lead to that.  Supposedly, it’s the extreme climate events as parturition and weaning can hardly have that effect. Please, amend.

I’m not convinced, though, that by simply adding “which can lead to a general scarcity of feed supply” the authors have answered Reviewer 1’s last comment (“The use of adequate nutrition… needs to be considered in a framework of general scarce feed supply). But it's up to Reviewer 1 to decide on this point.

Author Response

(The authors gave the same response as above.)

Round 3

Reviewer 3 Report

The paper has been improved by the authors.